# A cryptic third active site in cyanophycin synthetase creates primers for polymerization

Itai Sharon [1], Sharon Pinus[2], Marcel Grogg[3], Nicolas Moitessier [2], Donald Hilvert [3] &
T. Martin Schmeing [1✉]

Cyanophycin is a nitrogen reserve biopolymer in many bacteria that has promising industrial applications. Made by cyanophycin synthetase 1 (CphA1), it has a poly-L-Asp backbone with L-Arg residues attached to each aspartate sidechain. CphA1s are thought to typically require existing segments of cyanophycin to act as primers for cyanophycin polymerization. In this study, we show that most CphA1s will not require exogenous primers and discover the surprising cause of primer independence: CphA1 can make minute quantities of cyanophycin without primer, and an unexpected, cryptic metallopeptidase-like active site in the N-terminal domain of many CphA1s digests these into primers, solving the problem of primer availability. We present co-complex cryo-EM structures, make mutations that transition CphA1s between primer dependence and independence, and demonstrate that primer dependence can be a limiting factor for cyanophycin production in heterologous hosts. In CphA1, domains with opposite catalytic activities combine into a remarkable, self-sufficient, biosynthetic nanomachine.

[1] Department of Biochemistry and Centre de recherche en biologie structurale, McGill University, Montréal, QC H3G 0B1, Canada. [2] Department of Chemistry, McGill University, Montreal, QC H3A 0B8, Canada. [3] Laboratory of Organic Chemistry, ETH Zürich, CH-8093 Zürich, Switzerland.
✉email: martin.schmeing@mcgill.ca

Cyanophycin is a natural biopolymer discovered over 130 years ago as large, insoluble granules within cyanobacterial cells[1]. Cyanophycin chains consist of a poly-L-Asp backbone with L-Arg residues attached to each Asp side chain through isopeptide bonds[2]. Chains are typically ~80–400 β-Asp-Arg dipeptides in length $((\beta\text{-Asp-Arg})_{\sim 80-400})$[3,4]. The high nitrogen content and inert nature of cyanophycin make it ideal for storing fixed nitrogen[5], as well as carbon and energy[6,7]. Cyanophycin is especially useful for nitrogen-fixing cyanobacteria that separate aerobic photosynthesis and anaerobic nitrogen fixation either spatially or temporally[8,9], has been shown to enhance the efficiency of nitrogen assimilation in non-diazotrophic strains[10], and is also produced by many other bacteria across the kingdom[11]. Cyanophycin has promising potential commercial applications, from use as bandage material[12] to providing a source of poly-Asp, a biodegradable antiscalant, water softener, and super-swelling material[13]. Nevertheless, production yields of cyanophycin are currently too low for commercial viability and many studies have sought to increase them[14–17].

Cyanophycin synthetase 1 (CphA1) catalyzes polymerization of Asp and Arg into cyanophycin in two iterative, ATP-dependent reactions (Fig. 1a). In the first reaction, the C-terminus of a cyanophycin chain is activated by phosphorylation and then elongated by peptide bond formation with aspartic acid[18,19]. In the second reaction, the side-chain carboxylate of the newly added Asp residue is phosphorylated and then reacts with arginine to form an isopeptide bond. CphA1 contains dedicated domains for each reaction (Fig. 1b): the ATP-grasp family G domain ligates the Asp to the main chain, and the Mur ligase-like M domain adds the Arg to the Asp side chain. All CphA1 enzymes also contain an N-terminal domain (N domain), whose function was previously unknown. We recently showed that the N domain aids polymerization by loosely binding cyanophycin through charged patches, which helps the growing cyanophycin polymer alternate binding to G and M domain active sites[11,19]. Our study also visualized two separate tetrameric architectures for CphA1[11,19].

CphA1s have most often been described as possessing primer-dependent activity[3,19–21]. It is widely accepted that primer-dependent CphA1s cannot synthesize cyanophycin de novo from only Asp, Arg, and ATP, but require existing chains to extend. Only CphA1 from *Thermosynechococcus elongatus* BP-1 has been shown to display robust primer-independent activity[22]. Primer-dependent CphA1s are known to use long cyanophycin chains[20], trimer dipeptide segments $((\beta\text{-Asp-Arg})_3)$[18], and (albeit with low efficiency) other biomaterials[23] as primers, but characteristics of minimal and optimal primers are not established. It was also completely unknown what determines whether a CphA1 enzyme is primer dependent or primer independent. When heterologously expressed, CphA1 is catalytically active and produces cyanophycin within host cells[3,19–21,24–26], so understanding the nature of primers and primer independence could be important for bioproduction yields from these hosts.

Here, we report the discovery that the key to primer independence is a cryptic metallopeptidase-like active site in the N domain. We use a combination of cryo-EM, mass spectrometry, mutagenesis, and biochemical assays to characterize and

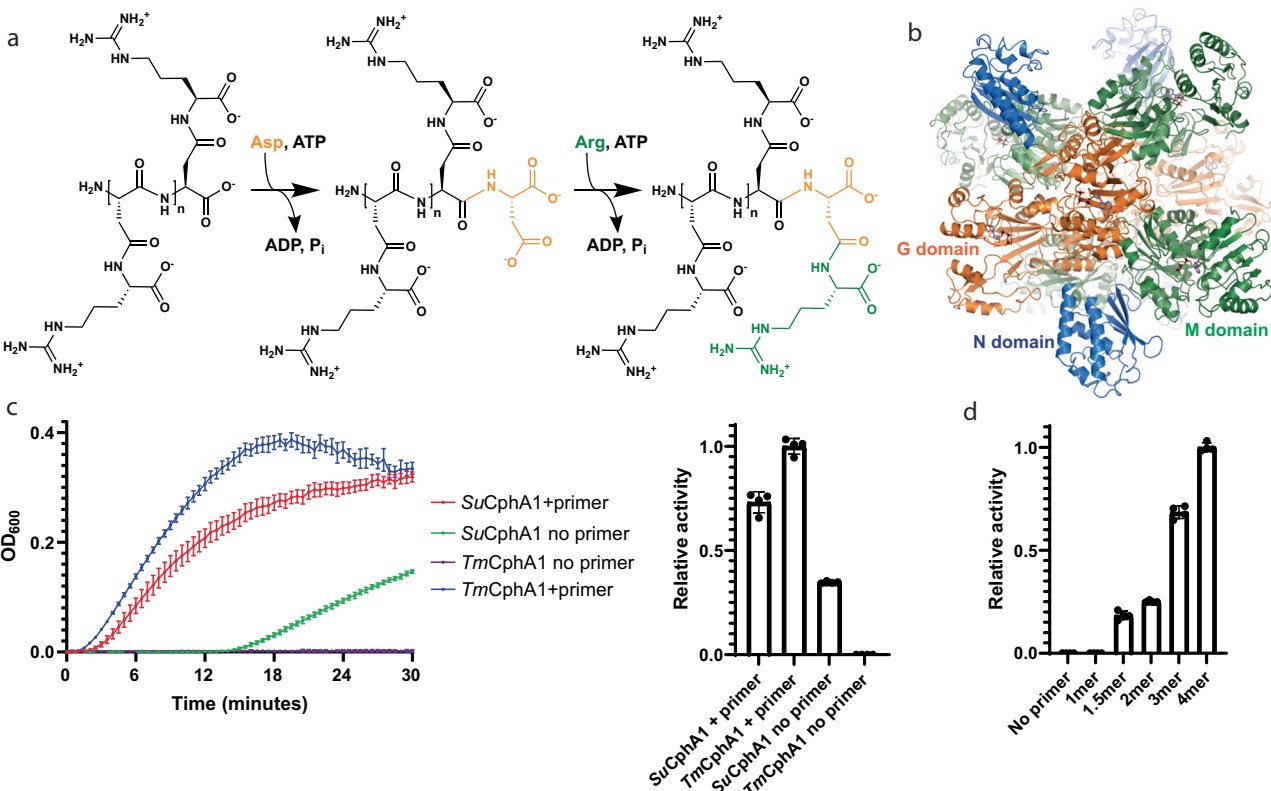

**Fig. 1 CphA1 structure and activity. a** Schematic diagram of the biosynthetic reactions catalyzed by the G and M domains of CphA1. **b** The overall structure of tetrameric CphA1[11] from *Synechocystis* sp. UTEX2470 (*Su*CphA1, PDB code 7LG5). ATP molecules mark the positions of the G (orange) and M (green) domain active sites. The N domain is colored in blue. **c** Cyanophycin biosynthesis plots and rate comparison of synthesis by *Su*CphA1 and *Tm*CphA1 with and without primer. *Tm*CphA1 is completely inactive in the absence of primer. $n = 4$ independent experiments. Data are presented as individual measurements and mean value, error bars represent SD values. **d** Activity levels of *Tm*CphA1 in the presence of various cyanophycin primers: 1mer (β-Asp-Arg)₁, 1.5mer (β-Asp-Arg)-Asp, 2mer (β-Asp-Arg)₂, 3mer (β-Asp-Arg)₃, 4mer (β-Asp-Arg)₄. $n = 4$ independent experiments. Data are presented as individual measurements and mean value, error bars represent SD values.

manipulate primer-independent CphA1 activity. The results show how the N domain enables biosynthesis by CphA1 without exogenous primers and demonstrate the implications of primer independence for in vivo cyanophycin production.

## Results

**Primer dependence of CphA1 enzymes.** Cyanophycin synthetases from *Synechocystis sp.* UTEX2470 (*Su*CphA1) and *Tatumella morbirosei* DSM23827 (*Tm*CphA1) show high activity in the presence of a ($\beta$-Asp-Arg)$_3$ primer[11,18,19] (Fig. 1c). Notably, in the absence of primer, *Su*CphA1 displays a lag phase of ~15 min, followed by robust cyanophycin synthesis. This result is surprising, because only one other CphA1 enzyme has ever been reported to be primer independent[22]. In contrast to *Su*CphA1, no primer-independent activity is observed for *Tm*CphA1, even at high protein concentrations and incubation with substrates over several days (Supplementary Fig. 1a). These results led us to ask what properties of the primer and enzyme control primer-dependent and -independent cyanophycin synthesis.

We first sought to define the minimal length of cyanophycin that can serve as a primer for CphA1 enzymes (Fig. 1d). *Tm*CphA1 could not perform cyanophycin synthesis in the presence of $\beta$-Asp-Arg, but the synthesis was observed with ($\beta$-Asp-Arg)$_2$, and the observed rate increases progressively in reactions with ($\beta$-Asp-Arg)$_3$ and ($\beta$-Asp-Arg)$_4$ (Fig. 1d). We then tested ($\beta$-Asp-Arg)-Asp, the CphA1 biosynthetic intermediate between $\beta$-Asp-Arg and ($\beta$-Asp-Arg)$_2$, and saw a similar activity as with ($\beta$-Asp-Arg)$_2$ (Fig. 1d). Assays with the primer-independent *Su*CphA1 showed similar results: Addition of ($\beta$-Asp-Arg)-Asp and longer fragments shortens the lag phase, and (Asp-Arg)$_3$ and ($\beta$-Asp-Arg)$_4$ give the highest rates of synthesis (Supplementary Fig. 1b). Together, these results define ($\beta$-Asp-Arg)-Asp as the minimal primer for CphA1.

**CphA1 primer independence does not depend on the G or M domain activity.** We next sought to identify the source of primer independence by comparing primer-independent *Su*CphA1 to primer-dependent *Tm*CphA1. We hypothesized that affinity differences for short cyanophycin segments at the G and/or M domain active sites might dictate primer (in)dependence[27]. *Su*CphA1 has several polar residues at the G and M domain active sites that could increase its affinity to nascent cyanophycin segments relative to *Tm*CphA1, which has hydrophobic residues at the analogous positions[11,19] (Supplementary Fig. 2a, b). However, experiments with 15 different mutants of *Tm*CphA1 in which one or more of these *Su*CphA1 hydrophilic residues were introduced into *Tm*CphA1's G domain, M domain, or G and M domains did not lead to primer independent activity (Supplementary Fig. 2c).

We then reasoned that if the G or M domains of *Su*CphA1 were individually responsible for primer independence, providing the necessary domain in trans to a reaction including *Tm*CphA1 would lead to primer-independent synthesis. Therefore, we prepared *Su*CphA1 with inactivating mutations[11] in the G domain (H267A), the M domain (D585A H586A), or in both (H267A; D585A H586A) (Supplementary Fig. 2a, b). To our surprise, all three of these constructs enabled robust primer-independent activity when added to *Tm*CphA1 (Fig. 2a). This result suggested that *Su*CphA1 has a third, previously unsuspected active site that is responsible for primer independence.

**A cryptic N domain active site responsible for primer independence.** Because our results indicated that the G and M domain active sites are not important for primer independence, we investigated whether the N domain is. We created a chimeric CphA1 (*Tm*CphA1$_{SuN}$) comprising the N domain of *Su*CphA1

and the G and M domains of *Tm*CphA1. Intriguingly, the *Tm*CphA1$_{SuN}$ chimera displayed robust primer-independent activity, suggesting that the N domain is responsible for primer independence (Fig. 2b). To verify this conclusion, we added the isolated *Su*CphA1 N domain in trans to a reaction including *Tm*CphA1. Again, primer-independent activity was observed, proving that the N domain is vital for primer independence in CphA1 (Fig. 2b and Supplementary Fig. 2d).

A catalytic role for the N domain has never been suggested before, so this discovery led us to re-examine the sequence conservation of N domains to search for a cryptic active site. No putative catalytic residues are 100% conserved, which is expected since some proportion of CphA1 enzymes will be primer dependent like *Tm*CphA1. However, an HxxEH motif[28] can be seen by careful inspection of sequence alignments (Fig. 2c and Supplementary Fig. 3a). The N domain of our existing *Su*CphA1 structures[11] shows the motif residues H79, E83, and H83 cluster together with C59, and are surrounded by several conserved arginine and histidine residues (Fig. 2d). All four residues of a Cx$_{19}$HxxEH motif are present in the primer-independent *Su*CphA1 and none of the four are present in the primer-dependent *Tm*CphA1 (Supplementary Fig. 3b). Of CphA1s with a non-redundancy of 70% (nr70), the Cx$_{19}$HxxEH motif is fully present in 83% of sequences. Mutations in and near this putative N domain active site (H57A, C59A, R70A, H79A, E82Q[29], and R100A) did not greatly affect primer-dependent activity (Supplementary Fig. 3c), but all except R100A reduced or abolished primer-independent activity (Fig. 2e), confirming that this N domain active site is responsible for primer independence. Notably, HxxEH is the active site motif for inverted zinc metallopeptidases[28], suggesting the N domain may have peptidase activity important for primer generation from cyanophycin polymer.

**The structural basis for the catalytic activity of the N domain.** To structurally characterize this cryptic N domain active site and its binding to cyanophycin, we turned to cryo-EM. We determined a structure of the inactivated[29] *Su*CphA1(E82Q) in complex with ($\beta$-Asp-Arg)$_{16}$ at 2.7 Å resolution by cryo-EM (Fig. 3a, b, Supplementary Fig. 4a, b, and Supplementary Table 1). The maps showed clear density for a chain of seven $\beta$-Asp-Arg dipeptide residues centered on the conserved region of the N domain. This region harbors a metallopeptidase-like active site[29] (Fig. 3c): H79 and H83 from the active site helix (residues 77–92) ligate an ion. We have tentatively assigned this ion as Zn$^{2+}$ because it is by far the most abundant metal detected in inductively coupled plasma mass spectrometry (ICP-MS) analyses of *Su*CphA1 (and not of *Tm*CphA1; Supplementary Table 2), and because of similarities to inverted zinc metallopeptidase[28] and peptide deformylase (PDF)[30] active sites, both of which can bind Zn$^{2+}$ tightly. The two histidine side chains bind Zn$^{2+}$ with their N$_\varepsilon$ atoms. H83, which forms π-stacking interactions with the conserved F67, also forms a hydrogen bond network with the conserved H57 and E87, stabilizing the tautomeric form in which its N$_\varepsilon$ lone-pair electrons face the Zn$^{2+}$ site[31]. C59, present in a loop region, serves as the third metal ligand, a role typically played by a glutamate, aspartate, or histidine from the metallopeptidase "glutamate helix"[29]. Q82 (taking the place of the general base E82 in this *Su*CphA1(E82Q) construct) sits above the metal-binding histidines, as in metallopeptidases. The cyanophycin polymer makes an extensive hydrogen-bonding network with itself and *Su*CphA1 residues (Fig. 3b), including the backbones of Y14, C59, A96, G97, and T101, and the side chains of E90, R70, Y110, S60 and M domain residues S603 and E607. These interactions position four visualized dipeptides upstream of the zinc ion and three downstream. The main chain carboxyl

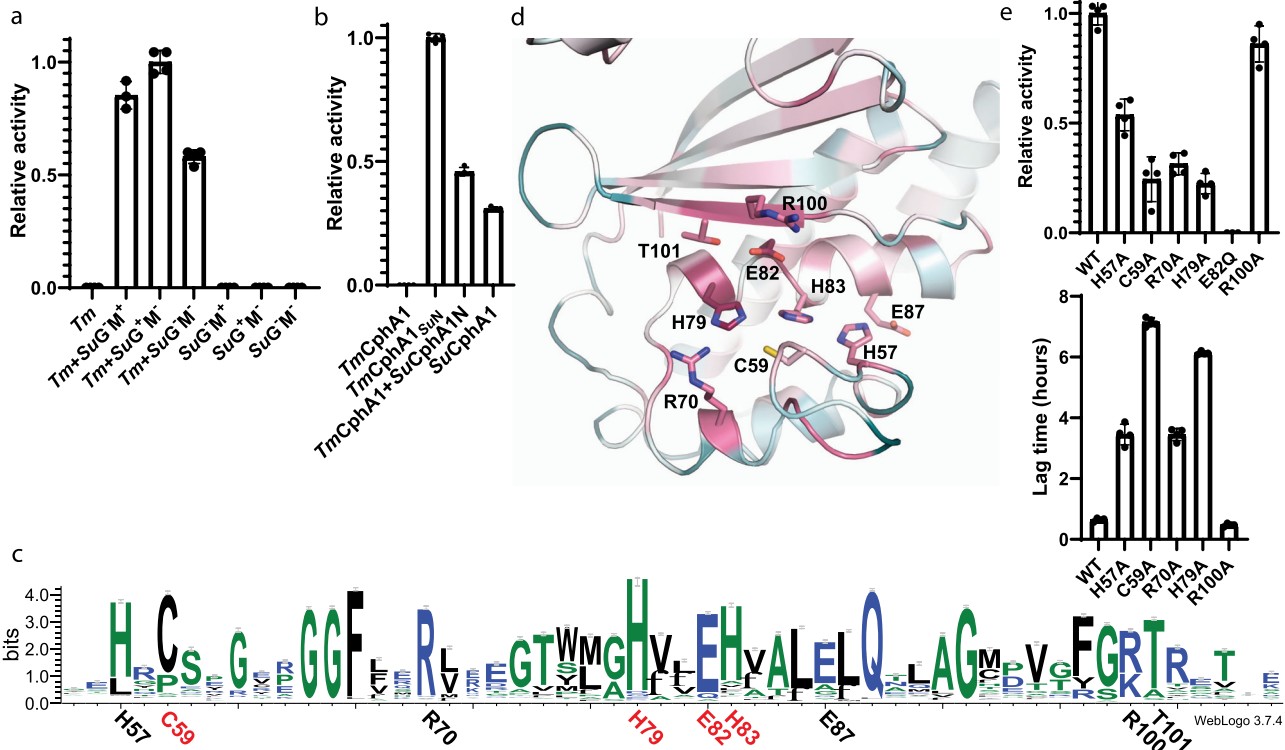

**Fig. 2 The N domain of CphA1 is responsible for primer-independent activity. a** Primer-independent activity of *Tm*CphA1 in the presence of *Su*CphA1 constructs with inactivating mutation as the G and/or M domains[11]. Active G and M domains are not required for activity, suggesting a different part of *Su*CphA1 is responsible for primer independence. $n = 4$ independent experiments. Data are presented as individual measurements and mean value, error bars represent SD values. **b** *Su*CphA1 N domain confers primer independence. *Tm*CphA1 is inactive in the absence of primers. A chimera of *Tm*CphA1 G and M domains and *Su*CphA1 N domain (*Tm*CphA1$_{SuN}$) is active. Reactions including *Tm*CphA1 and 5 μM of a construct of the extruded N domain of *Su*CphA1 harboring solubilizing mutations Y14S and I17T show cyanophycin synthesis in the absence of primers. Note that the extruded *Su*CphA1 N domain was completely insoluble before introducing mutations into a hydrophobic loop that, in the intact enzyme, interacts with the M domain (Supplementary Fig. 2d). $n = 4$ independent experiments. Data are presented as individual measurements and mean value, error bars represent SD values. **c** Weblogo[60] diagram showing the conserved Cx$_{19}$HxxEH motif. This Weblogo was constructed from sequence alignments of all CphA1 enzymes generated by ClustalOmega[61], and excludes cyanophycin synthetase 2 (CphA2) sequences. CphA2s are specialized cyanobacterial enzymes that polymerize β-Asp-Arg dipeptides recovered from degraded cyanophycin[27,62]. CphA2 N domains share a low sequence identity with CphA1 N domains and the N domain active site motif is absent from CphA2 sequences. **d** The N domain of *Su*CphA1 colored by per-residue conservation. The residues around the Cx$_{19}$HxxEH motif are conserved (purple). The conservation was calculated by Consurf[63] using 500 randomly chosen CphA1 sequences. **e** Activity rate and lag time (time until $V_{max}$ is reached) of *Su*CphA1 N domain mutants without added primer. All mutants except R100A displayed varying levels of reduced activity rate and longer lag phases than the WT enzyme. E82Q was inactive and its lag time is not shown. $n = 4$ independent experiments. Data are presented as individual measurements and mean value, error bars represent SD values.

oxygen of the fourth dipeptide is 2.6 Å from Q82 and 3.6 Å from the Zn$^{2+}$ ion, in a good pre-cleavage position.

We had also previously calculated a cryo-EM map of wildtype (WT) *Su*CphA1 in the presence of (β-Asp-Arg)$_{16}$[11]. The new results described here led us to re-examine it. Signal consistent with cyanophycin bound to the N domain is visible in that map, although not quite as strong as that seen with *Su*CphA1(E82Q) (Fig. 3d, Supplementary Fig. 4c, and Supplementary Table 1). This signal is not present in maps of *Su*CphA1 that was not incubated with cyanophycin segments[11]. Interestingly, we were able to fit the first four β-Asp-Arg dipeptide residues into this map in the same conformation as seen bound to *Su*C-phA1(E82Q), but there is no signal for any dipeptide residues C-terminal to the N domain active site. The C-terminal Asp carboxyl group is positioned directly next to the Zn$^{2+}$ ion, indicating this represents an N domain product complex, derived from in situ cleavage of (β-Asp-Arg)$_{16}$.

**The N domain cleaves cyanophycin into primers.** The structures suggest that the N domain possesses endo-cyanophycinase activity.

To directly observe this activity, we incubated purified cyanophycin with *Su*CphA1 and examined the reaction with SDS-PAGE (Supplementary Figs. 5a and 7). A slow but clear decrease in cyanophycin is observed over several days, especially in the molecular weight range of ~15–20 kDa. We next performed mass spectrometry-based cyanophycin cleavage assays. Incubation of *Su*CphA1 with (β-Asp-Arg)$_8$-NH$_2$ over several hours led to the gradual formation of species with mass values corresponding to (β-Asp-Arg)$_4$-NH$_2$ and (β-Asp-Arg)$_4$ (Fig. 4a). Likewise, (β-Asp-Arg)$_8$-Asn is converted by *Su*CphA1 to species with masses corresponding to (β-Asp-Arg)$_4$-Asn and (β-Asp-Arg)$_4$ (Fig. 4b). Control reactions with primer-dependent *Tm*CphA1 or the N domain *Su*C-phA1(E82Q) variant with (β-Asp-Arg)$_8$-NH$_2$ or (β-Asp-Arg)$_8$-Asn did not result in the appearance of the product peaks (Supplementary Fig. 5b). *Su*CphA1 cleaved (β-Asp-Arg)$_{12}$ into major products (β-Asp-Arg)$_8$ and (β-Asp-Arg)$_4$, and minor products (β-Asp-Arg)$_5$ and (β-Asp-Arg)$_7$ (Supplementary Fig. 5c–e). Thus, the N domain of *Su*CphA1 is indeed a cryptic primer-generating endo-cyanophycinase that possesses a low catalytic rate and, at least with the cyanophycin segments used in these experiments, preferentially yields (β-Asp-Arg)$_4$ fragments as products.

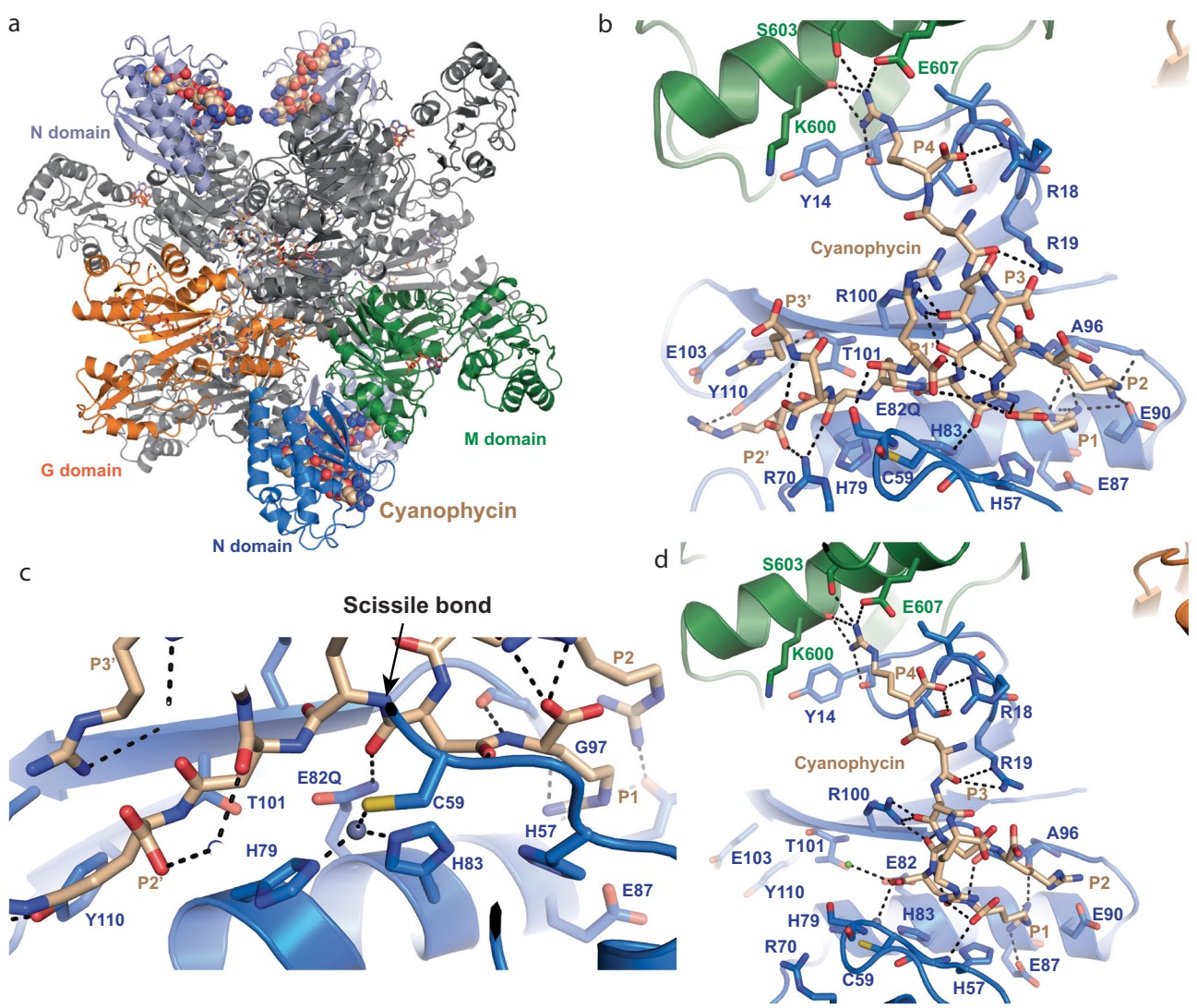

**Fig. 3 The structure of SuCphA1 N domain with bound cyanophycin. a** Structure of *Su*CphA1 in complex with cyanophycin substrate in both the G domain active site (sticks) and N domain active site (spheres). **b** The *Su*CphA1 N domain in complex with (β-Asp-Arg)₁₆ as a substrate. Seven dipeptide residues are visible. Polymer binding residues and their interactions are highlighted. P4, P3, P2, P1, P1′, P2′, P3′ denote β-Asp-Arg dipeptides numbered relative to the cleavage point. **c** Close-up view of the structure of *Su*CphA1 in complex with cyanophycin substrate. **d** The structure of *Su*CphA1 in complex with in situ cleaved cyanophycin. Four dipeptide residues are visible. Polymer binding residues and their interactions are highlighted.

The relative geometry of the CphA1 active sites is important for cyanophycin biosynthesis[11], so we interrogated whether it is also important for primer-generating cleavage. We combined the W672A mutation (which forces *Su*CphA1 to be dimeric instead of tetrameric[11]) with mutations that abolish the activity of each active site (E82Q = N⁻; H267A = G⁻; D585A H586A = M⁻) and used orthogonal affinity tags to purify different *Su*CphA1 heterodimers[11]. *Su*CphA1 N⁺G⁺M⁺/N⁻G⁻M⁻ (with all WT active sites on the same protomer) and *Su*CphA1 N⁻G⁺M⁺/N⁺G⁻M⁻ (with the WT N domain on the opposite protomer as WT G and M domains) have similar activity in the presence of primer (Fig. 4c). However, N⁻G⁺M⁺/N⁺G⁻M⁻ is somewhat less active in the absence of primer (Fig. 4d). This suggests that the proximity of the hydrolytic active site to the biosynthetic active sites is beneficial for primer-independent activity. Interestingly, *Su*CphA1 dimers[11,19] exhibit the same primer-dependent activity rate as tetramers, but a lower primer-independent activity rate (Fig. 4e), indicating that the tetramer architecture of *Su*CphA1 is also beneficial for primer-independent activity.

**Effect of primer independence on heterologous cyanophycin synthesis.** Understanding the basis of primer dependence in cyanophycin biosynthesis and obtaining primer-independent and primer-dependent variants of the same CphA1 enzymes allowed us to examine the importance of primer availability for cyanophycin accumulation in a heterologous host. To that end, we separately expressed primer-independent *Su*CphA1 and primer-dependent *Su*CphA1(E82Q) in *Escherichia coli* BL21(DE3) and quantified the amount of polymer produced in vivo by each variant. *E. coli* harboring WT *Su*CphA1 produced on average 2.3-fold more cyanophycin than the E82Q mutant, measured as milligrams of cyanophycin per liter of growth culture (Table 1). The total wet cell mass was 12% lower for cells expressing the WT enzyme, suggesting they divert more resources to cyanophycin synthesis from cell growth compared to those harboring the primer-dependent mutant. Similarly, in experiments with primer-dependent *Tm*CphA1 and the primer-independent chimera *Tm*CphA1_SuN, the chimera produced 2-fold more cyanophycin than the primer-dependent WT enzyme, in a lower total wet cell mass (Table 1).

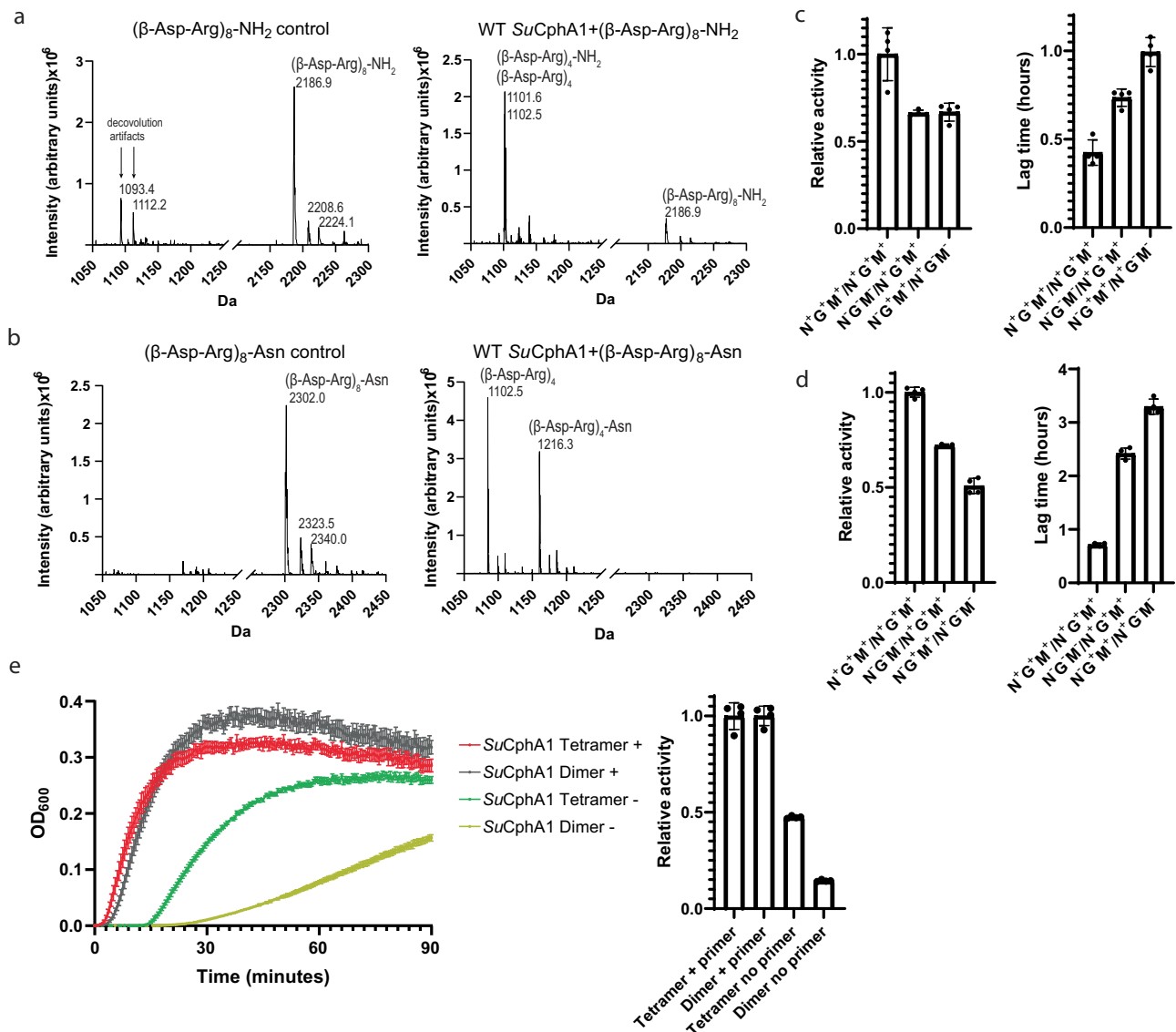

**Fig. 4 Catalytic activity of the N domain and cyanophycin synthesis by dimeric CphA1. a** Mass spectra of (β-Asp-Arg)$_8$-NH$_2$, an 8mer cyanophycin segment in which the terminal carboxylate is replaced by an amide, before (left) and after (right) incubation with WT SuCphA1. After incubation with enzyme, the peak corresponding to (β-Asp-Arg)$_8$ (expected at 2187.0 Da) is reduced, and two peaks with sizes matching to (β-Asp-Arg)$_4$ (expected at 1102.5 Da) and (β-Asp-Arg)$_4$-NH$_2$ (expected at 1101.5 Da) appear. Peaks corresponding to Na$^+$ and K$^+$ adducts are also labeled. Peaks at 1093.4 and 1112.2 Da are deconvolution artifacts (Supplementary Fig. 5). **b** Mass spectra of (β-Asp-Arg)$_8$-Asn before (left) and after (right) incubation with WT SuCphA1. After incubation with enzyme, the peak corresponding to (β-Asp-Arg)$_8$-Asn (expected at 2301.1 Da) disappears and two peaks with sizes matching to (β-Asp-Arg)$_4$ (expected at 1102.5 Da) and (β-Asp-Arg)$_4$-Asn (expected at 1216.6 Da) appear. **c**, **d** Activity rate and lag time of SuCphA1 dimer complementation assays with **c** and without **d** primer. In presence of exogenous primer, the two mutant combinations G$^-$M$^-$N$^-$/G$^+$M$^+$N$^+$ and G$^-$M$^+$N$^+$/G$^+$M$^-$N$^-$ display a similar activity rate, although the G$^-$M$^+$N$^+$/G$^+$M$^-$N$^-$ combination displays a somewhat longer lag time. In the absence of primer, the G$^-$M$^-$N$^-$/G$^+$M$^+$N$^+$ combination displays a somewhat higher activity rate and lower lag time. $n = 4$ independent experiments. Data are presented as individual measurements and mean value, error bars represent SD values. **e** Activity assays and activity rate comparison of WT SuCphA1 (tetramer) and the W672A mutant (dimer) with (+) and without (−) primer. The two enzymes display similar primer-dependent activity, but the dimer has lower primer-independent activity. $n = 4$ independent experiments. Data are presented as individual measurements and mean value, error bars represent SD values.

| Table 1 Heterologous cyanophycin production in *E. coli*. | | | |
|---|---|---|---|
| **Enzyme** | **Cell pellet (mg)** | **Polymer (mg)** | **% of cell mass (w/w)** |
| *Su*CphA1 WT | 4730 ± 160 | 413 ± 35 | 8.7 |
| *Su*CphA1 E82Q | 5390 ± 140 | 177 ± 28 | 3.2 |
| *Tm*CphA1 WT | 5570 ± 380 | 247 ± 18 | 4.4 |
| *Tm*CphA1$_{SuN}$ | 4880 ± 170 | 486 ± 52 | 9.9 |

## Discussion

The sequence, structure, and activity data all indicate that the N domain of CphA1 has cryptic metallopeptidase activity. The endo-cyanophicinase activity of the N domain presents an explanation for the primer dependence and independence in CphA1 enzymes: We propose that all CphA1 enzymes possess very low levels of true primer-independent activity for the first steps of cyanophycin synthesis, e.g., ligating Asp and Arg to β-Asp-Arg, and ligating β-Asp-Arg and Asp to (β-Asp-Arg)-Asp. In

the next steps of elongation of these intermediates, the rate of polymerization increases, and a long chain of cyanophycin is made. In CphA1 enzymes with N domain metallopeptidase activity, the chain is cleaved to generate cyanophycin segments such as (β-Asp-Arg)$_4$ that act efficiently as primers. This leads to more long chains and more primers, and thus rapid accumulation of cyanophycin after the initial lag phase we observe. CphA1 enzymes that lack active N domains make long cyanophycin chains as well, but because they are limited by the very slow initial rates in absence of primers, they make so few as to be undetectable in light-scattering or ATP hydrolysis assays[3,19–21] (Supplementary Fig. 1a).

In vivo, these CphA1 enzymes that do not have N domain active sites likely use remnant strands of cyanophycin left over from the last round of catabolism, or other cellular small molecules[23] as a primer. The maximum rates we observe in vitro indicate that polymerization of cyanophycin is several-fold faster than hydrolytic cleavage, but it is difficult to relate these rates to the situation in vivo, where cellular conditions are not constant and the availability of cyanophycin chains will change as molecules aggregate into granules. However, the accumulation of large amounts of cyanophycin in native bacteria and heterologous hosts clearly indicates the relative in vivo rates of polymerization and hydrolytic primer production are well tuned for cyanophycin biosynthesis.

Intriguingly, our experiments with dimeric CphA1 mutants (Fig. 4) suggest that a nascent chain could be polymerized at one end while being cleaved near the other end. In the absence of exogenous primer, mutant heterodimeric SuCphA1 displays higher synthesis rates when all three intact active sites are in the same protomer, hinting the increased rate is from cleavage in cis. A cyanophycin chain being elongated at its C-terminus by the G and M domains with soft anchoring on the N domain helices α$_a$ and α$_b$ could intermittently wrap around to be cleaved at that same N domain's active site (Supplementary Fig. 6a). Similarly, geometrical considerations can rationalize why dimeric SuCphA1 is less active than tetrameric SuCphA1 in the absence of exogenous primer: In the tetramer, two N domains active sites face each other and are 55 Å apart (protomers A and C; Supplementary Fig. 6a). It is possible that after cleavage, the new N-terminus of a cyanophycin chain experiences increased local concentration of N domain active sites, facilitating binding and increasing the rate of hydrolytic primer production. We note that the primer-dependent TmCphA1 has a different tetramer architecture[11], in which the equivalent N domain positions are ~80 Å apart (Supplementary Fig. 6b).

The N domain appears distantly related to the M16 peptidase family[32,33], which includes endopeptidases such as pitrilysin[29] and insulin-degrading enzyme[34]. The family is also known as inverzincins[35] because the active site motif HxxEH is inverted from the HExxH of the canonical mononuclear metallopeptidase motif. Active CphA1 N domains share the inverzincin[35] HxxEH motif, as well as three structural elements: an active site helix, an adjacent β sheet, and the "backing helix"[35] (Supplementary Fig. 6c). The CphA1 backing helix doubles as the α$_a$ helix, which binds nascent cyanophycin chains through its surface-exposed side during biosynthesis[11]. Substrate binding in CphA1 and pitrilysin[29] is similar, with the scissile peptide bond in analogous positions (Supplementary Fig. 6d). However, CphA1 N domains are clearly distinct from known inverzincins: CphA1 N domains are much smaller (~160 residues vs. up to 1000 residues), the structural similarity is modest and confined to the region around the active site, and crucially, the third metal-binding residue in CphA1 is a Cys upstream in sequence from the histidines in a C-H-H metal-binding triad, rather than a downstream Asp, Glu or His in an M16 peptidase H-H-D/E/H triad[31,35,36].

Two or more Cys ligands are common in structural Zn$^{2+}$-binding motifs, but Cys as a Zn$^{2+}$ ligand in an active metallopeptidase is rare[35,37–39]. The best-known example of a metallopeptidase with C-H-H metal coordinating residues is PDF, a ubiquitous enzyme responsible for deformylation of N-terminal fMet residues[30]. CphA1 N domains and PDFs share a very little structural similarity, and their active site helices are in opposite orientations (Supplementary Fig. 6e), but the geometry of the metal-binding residues of the two enzymes are remarkably similar (Supplementary Fig. 6f)[40]. PDF can bind Zn$^{2+}$ tightly, but has a lower catalytic rate when bound with zinc than when bound with cobalt, nickel[41], or iron[42]. The peptidase activity of CphA1 needs to be properly tuned so hydrolysis can generate primers but not efficiently compete with polymerization in this biosynthetic enzyme tasked with making long cyanophycin chains for storage. Because the biosynthetic and hydrolytic activities are both encoded into the same enzyme, the balance of these activities cannot be regulated by protein expression levels. Other features of the N domain active site that may temper the rate of hydrolysis are the lack of a residue for transition state stabilization (inverted zinc metallopeptidase[28] has a Tyr; PDF[43] has a Gln), or the lack of an active site residue accepting a hydrogen bond from N$_\delta$ of H79. The latter would promote the N$_\epsilon$ lone-pair electrons facing the metal-binding site that can be important for activity[31,44]. H83 has such an interaction, but it is common for it to be seen for both active site histidines[31,44].

Long strands of cyanophycin precipitate into granules for storage[4,45]. This precipitation may also serve to sequester cyanophycin from CphA1's hydrolytic activity, since these chains of cyanophycin are largely stable both in vivo and in vitro in the presence of CphA1s which have active N domains[19,22,45] (Supplementary Fig. 5a). Similarly, sequestration from the polymerizing G and M domains by precipitation into multistrand granules may be involved in determining cyanophycin chain length, which varies with CphA1, host, and other factors[19,20,24,45,46]. The exo-cyanophycinase CphB has a high $V_{max}$ and an active site that is shallow and accessible[47,48], allowing rapid degradation of strands in granules when needed.

The in vivo experiments we performed show that primer dependence can be a limiting factor for cyanophycin production in heterologous hosts. This understanding can help guide future efforts for more efficient polymer production in vivo, for example by prioritizing primer-independent enzymes. With ~80% of CphA1s having the Cx$_{19}$HxxEH motif, primer-independent CphA1s are more common than previously realized, and primer independence can be conferred by using N domain chimeras like TmCphA1$_{SuN}$. Of the four constructs we assayed here, the chimera that introduces an active N domain into a primer-dependent CphA1 produced the highest cyanophycin yields.

Although cryptic active sites are not unheard of[49], it is unusual to discover a cryptic active site in an enzyme that has been studied for decades. However, it was completely unexpected that cyanophycin synthetase would generate its own primers or have hydrolytic activity for any reason, given its biosynthetic role. Furthermore, the active site motif was obscured in sequence alignments (Fig. 2c and Supplementary Fig. 3a) by the ~20% of primer-dependent CphA1 enzymes that do not have the active site, and the structural similarity to metalloproteases is so modest that they do not appear in the top 100 results of DALI[50] searches. Only the observation that the N domain confers primer independence led to the discovery of the N domain cyanophycinase site, thus showing that evolution combined three different enzymes into one elegant macromolecular machine. We are not aware of any other polymerase that has a dedicated active site to create primers needed for its biosynthetic cycle, making CphA1 a truly remarkable, multifunctional enzyme.

## Methods

**Cloning, protein expression, and purification**. The genes encoding *Su*CphA1 (protein WP_028947105.1) and *Tm*CphA1 (protein WP_004925893.1) were cloned into pJ411-derived plasmids in a previous study[11]. Point mutants and (sub) domain chimeras used in this study were generated by transforming DH5-α *E. coli* cells with PCR fragments containing overlapping ends. Phusion® DNA polymerase (New England Biolabs) was used for all PCR reactions. Sequences of DNA primers used for cloning are listed in Supplementary Table 3. Proteins were expressed in *E. coli* BL21(DE3). Cells were grown in LB media supplemented with 100 µg/ml kanamycin at 37 °C until OD600 reached ~0.5. The growth temperature was then reduced to 22 °C and protein expression was induced with 0.25 mM isopropyl β-d-1-thiogalactopyranoside (IPTG) for ~20 h. Following harvesting by centrifugation, the cells were resuspended in buffer A (250 mM NaCl, 50 mM Tris pH 8.0, 10 mM imidazole, 2 mM β-mercaptoethanol) supplemented with a few crystals of lysozyme and DNAse I, and lysed by sonication at 0 °C. The lysate was clarified by centrifugation at 40,000 g, loaded onto a HisTrap HP column (Cytiva), washed extensively with buffer B (buffer A with 30 mM imidazole), and eluted with buffer C (buffer A with 250 mM imidazole). For structural studies, the proteins were incubated with TEV protease for removal of the affinity tag while being dialyzed overnight against buffer D (250 mM NaCl, 20 mM Tris pH 8, 5 mM β-mercaptoethanol) and then applied again to a HisTrap column. Resulting samples were concentrated using Amicon centrifugation concentrators (EMD Millipore) and loaded onto a Superdex200 16/60 column (GE Healthcare) equilibrated in buffer E (100 mM NaCl, 20 mM Tris pH 8.0, 1 mM dithiothreitol). Following gel filtration, fractions with the highest purity were pooled and concentrated to ~20 mg/ml. Glycerol was added to a final concentration of 10% v/v, and the samples were flash frozen and stored at −80 °C until use.

For dimer complementation experiments, *Su*CphA1_W672A carrying the desired active-site mutations was cloned into pCDF-derived plasmids with a C-terminal calmodulin-binding protein tag. *E. coli* BL21(DE3) cells were co-transformed with a pJ411-derived plasmid (for a His-tagged version) and a pCDF-derived plasmid and grown in LB media supplemented with 100 µg/ml kanamycin and 100 µg/ml spectinomycin as described above. All purification steps were similar to those already described up to the elution from the HisTrap HP column. Following elution, the protein was mixed with CaCl₂ to a final concentration of 2 mM and loaded onto a column of calmodulin-sepharose (Agilent) equilibrated with buffer F (250 mM NaCl, 50 mM Tris pH 8.0, 2 mM CaCl₂, 2 mM β-mercaptoethanol), washed with buffer F and eluted with buffer G (250 mM NaCl, 50 mM Tris pH 8.0, 2 mM EGTA, 2 mM β-mercaptoethanol). The eluted protein was buffer exchanged into buffer E, concentrated and frozen.

**Cryo-EM grid preparation, data collection, and processing**. *Su*CphA1(E82Q) (3.5 mg/ml) was mixed with 2 mM ATP, 10 mM MgCl₂, 1 mM (β-Asp-Arg)₁₆ and 0.09% octyl β-D-glucopyranoside. Three microliters of this sample were applied to glow-discharged C-flat 300 mesh 1.2/1.3 Cu holey carbon grids, blotted for 3 s at 4 °C and 90% humidity using a Vitrobot IV (FEI) and plunge-frozen into liquid ethane. Data were collected at the McGill Facility for EM Research using an FEI Titan Krios TEM operating at 300 kV with a Gatan K3 DED and a Gatan GIF BioQuantum LS. Movies were collected in counting mode using SerialEM, with a total dose of 60 e/Å² over 30 frames and a set defocus range of −1.0 to −2.0 µm at a nominal magnification of 105,000, resulting in a pixel size of 0.855 Å². Micrographs were motion corrected using Relion3.1[51]. The motion-corrected micrographs were imported to CryoSPARC2[52] for patch-CTF estimation, particle picking, and several rounds of 2D and 3D classification to remove junk particles. The particles were then exported to Relion3.1[51] for 3D refinement followed by two rounds of Bayesian polishing and CTF refinement. The polished particles were then exported to CryoSPARC2 and 3D refined using homogeneous refinement with defocus and high-order aberrations refinement. Local resolution estimation followed by local filtering was then performed in CryoSPARC2, and the locally filtered map was used for model building. The map of WT *Su*CphA1 with ATP and (β-Asp-Arg)₁₆ was calculated in a previous study[11] and deposited as EMDB-23326.

**Structure refinement**. The previously determined structure of *Su*CphA1 with ATP (PDB 7LG5) was used as a starting model for the two structures. The model was manually docked into the maps using UCSF Chimera[53] and refined using Rosetta[54]. Further refinement of the protein and positioning of the substrate molecules were done manually in Coot[55], using the model validation feature in CCP-EM 1.4[56] for guidance. Conformational constraints of substrates were generated in CCP4i2[57]. Figures were generated using PyMOL.

**CphA1 activity assays**. CphA1 activity was monitored by following scattering of light by cyanophicin at neutral pH as previously described[11]. Unless stated otherwise, reactions contained 700 nM purified CphA1, 100 mM HEPES pH 8.2, 20 mM KCl, 10 mM MgCl₂, 2 mM each L-Asp and L-Arg, 4 mM ATP, and 50 µM synthetic cyanophicin primer as indicated. The reaction volume was 100 µl and reactions were performed in quadruplicate. OD600 was monitored using a SpectraMax Paradigm spectrophotometer running SoftMax Pro 5.4.1 (Molecular Devices), with 5 s linear shaking between reads. Data were analyzed using GraphPad Prism. To calculate maximal rates, the maximum of the first derivative

of each OD600 curve was taken. The derivatives curves were smoothed with a second-order polynomial to reduce noise in measurements. Lag time to maximal rate is the time when the first derivative reaches its maximal value.

**Cyanophicin purification**. *E. coli* BL21(DE3) cells were transformed with the same plasmids used for protein expression and plated on LB plates supplemented with 50 µg/ml kanamycin. The next day, single colonies were picked and used to inoculate 10 ml of LB supplemented with 100 µg/ml kanamycin. The starter culture was grown overnight with shaking at 37 °C and then used to inoculate 1 l of LB supplemented with 100 µg/ml kanamycin. One-liter cultures were grown with shaking at 37 °C until OD600 reached 0.5, and then the temperature was reduced to 25 °C. After 1 h, protein expression was induced with 0.25 mM IPTG, and the cultures grown for another 20 h. The next day, cells were harvested by centrifugation, resuspended in 1 ml ddH₂O for every 0.2 g of cell pellet, and lysed by sonication at room temperature. The lysates were acidified to pH 0.9 using concentrated HCl and clarified by centrifugation at 3500 g for 20 min. The pH of the clarified lysate was then neutralized using 2 M NaOH. Following centrifugation at 3500 g for 10 min, the pellets contained insoluble cyanophicin and the lysate contained soluble cyanophicin. The pellets were resuspended in 0.1 M HCl, centrifuged at 3500 g for 10 min and the resulting pellets were discarded. The pH of the liquid phase was neutralized with 2 M NaOH and centrifuged for 10 min at 3500 g. The resulting pellets, consisting of purified insoluble cyanophicin, were lyophilized and weighed. The lysate containing soluble cyanophicin was mixed with 1 volume of 95% EtOH and centrifuged for 10 min at 3500 g. The resulting pellet was resuspended in ddH₂O, mixed with 1 volume of 95% EtOH, and centrifuged for 10 min at 3500 g. The resulting pellet, consisting of purified soluble cyanophicin, was lyophilized and weighed. The reported amounts of purified cyanophicin are the sum of the soluble and insoluble polymer from each culture.

**MS analysis of cyanophicin degradation**. Synthetic cyanophicin segments were digested in 100 µl reactions containing 1 µM purified CphA1, 100 mM (NH₄)₂CO₃, 20 mM KCl and 5 mM MgCl₂, and 2 mM cyanophicin segments. Samples of 10 µl were taken at specific time points and diluted into 90 µl of 100 mM (NH₄)₂CO₃, then directly injected for 2 min at 40 µl/min into a Bruker amaZon speed ETD ion trap mass spectrometer operating at positive ionization mode. The resulting spectra were deconvoluted using the max entropy method.

**SDS-PAGE analysis of cyanophicin degradation**. Reactions contained 20 µM purified CphA1, 50 mM (NH₄)₂CO₃, 20 mM KCl and 5 mM MgCl₂, 5 mg/ml cyanophicin purified from *E. coli*, 0.01% NaN₃ and 200 µM phenylmethylsulfonyl fluoride. The reactions were incubated at room temperature. Samples of 20 µl were mixed with 10 µl of 5× loading buffer, boiled for 1 min, and analyzed on a 17% polyacrylamide gel.

**Synthesis of cyanophicin segments**. (β-Asp-Arg) dipeptides were made from purified cyanophicin made in vitro in a primer-independent reaction by *Su*CphA1. The produced polymer was isolated by centrifugation at 3500 g for 10 min, washed with ddH₂O, and resuspended in 50 mM (NH₄)₂CO₃. The polymer suspension was digested with purified cyanophicinase from *Synechocystis sp.* PCC6803[48] until the suspension became clear, then filtered using a 3 kDa molecular weight cut-off Amicon centrifugation concentrator (EMD Millipore) and lyophilized.

All other cyanophicin segments were prepared by manual Fmoc solid-phase peptide synthesis (SPPS) as previously described[11,58,59]. Briefly, (β-Asp-Arg)ₙ, where *n* = 2, 3, 4, 8, or 16, were synthesized on an HMPB-ChemMatrix resin (Biotage) on a 0.01–0.03 mmol scale using Fmoc-(β-Asp-Arg)(Pbf)-OH as the building block. Fmoc groups on the growing chains were removed with piperidine in DMF, and coupling was carried out with HATU/DIPEA in DMF. Cleavage of the peptides from the resin and removal of the O*t*Bu and Pbf protecting groups were achieved with TFA-H₂0-ⁱPrSiH (95:2.5:2.5). (β-Asp-Arg)-Asp and (β-Asp-Arg)ₙ-Asn (*n* = 4 and 8) were prepared analogously, but use Fmoc-Asp (O*t*Bu)-OH or Fmoc-Asn(Trt)-OH rather than Fmoc-(β-Asp-Arg)(O*t*Bu,Pbf)- OH for the first coupling to the resin; the allyl protecting group was removed with Pd(PPh₃)₄ and PhSiH₃ in the final deprotection step. The C-terminal amides (β-Asp-Arg)ₙ-NH₂ (*n* = 4 and 8) were synthesized by manual Fmoc-SPPS on an N-alkylated PAL resin (Bachem); couplings and peptide release from the resin were otherwise the same as for the other derivatives. All products were purified by reverse phase preparative HPLC and analyzed by high-resolution mass spectrometry[11] (Supplementary Table 4).

**Metal analysis**. For metal analysis, purified protein samples of *Su*CphA1, its extruded N domain, and *Tm*CphA1 were buffer exchanged into 100 mM (NH₄)₂CO₃ by performing gel filtration with a Superdex S200 10/300 column equilibrated with that buffer. Protein-containing fractions were concentrated to 100 µM and analyzed by ICP-MS at the Center for Applied Isotope Studies, University of Georgia. A sample of the buffer eluted from the column was used as a control.

**Reporting summary**. Further information on research design is available in the Nature Research Reporting Summary linked to this article.

## Data availability

The structural models and maps (Supplementary Fig. 4 and Supplementary Table 1) generated in this study are available in the Protein Data Bank database under accession codes 7TXU and 7TXV and the Electron Microscopy Data Bank under accession code EMD-26161. The biochemical data (Figs. 1c, d, 2a, b, e, and 4, Table 1, Supplementary Figs. 1, 2c, 3c, and 5a–e, and Supplementary Table 4) generated in this study are provided in the source data file. Source data are provided with this paper.

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

## Acknowledgements

We thank all the members of the Schmeing lab for advice and ongoing discussions on this project, J.F. Trempe for advice with mass spectrometry, Christopher Thibodeaux and Kenneth Johnson for advice with data interpretation, Nancy Rogerson for proofreading, staff at McGill Facility of EM Research (Kaustuv Basu and Kelly Sears) for support during data collection and the Plasma Chemistry Laboratory at the Center for Applied Isotope Studies, University of Georgia for ICP-MS. The work (10.46936/10.25585/60001153) conducted by the U.S. Department of Energy Joint Genome Institute (https://ror.org/04xm1d337), a DOE Office of Science User Facility, is supported by the Office of Science of the U.S. Department of Energy operated under Contract No. DE-AC02-05CH11231. This work was funded by CIHR Project Grant 178084 and a Canada Research Chair to T.M.S., and the Schweizerischer Nationalfonds and ETH Zurich to D.H.

## Author contributions

M.G. and S.P. performed the chemical synthesis of cyanophycin segments. I.S. performed all other biochemical and structural experiments and data processing. I.S. and T.M.S. wrote the manuscript with editing from D.H. N.S. supervised S.P.

## Competing interests

The authors declare no competing interests.
