## [Peer Review File · Nature Communications]

REVIEWER COMMENTS

Reviewer #1 (Remarks to the Author):

The paper „A cryptic third active site in cyanophycin synthetase creates primers for polymerization“ by Sharon et al is excellent work. It is a follow up of the recently resolved structure and detailed reaction mechanism of the unique enzyme Cyanophycin synthetase (CphA1), which produces the polymer cyanophycin (CP), which is multi-L-arginyl-poly-L-aspartate, a nitrogen rich reserved polymer ubiquitously found in cyanobacteria and beyond. The work starts with the observation that the CphA1 enzyme from *Thermosynechococcus* starts to produce CP after a lag phase of 15 min independent of the addition of short CP primers while another CP required the addition of at least an (Asp-Arg)-dimer as primer. In a series of elegant experiments the authors identified the N-domain of the enzyme to be responsible for primer formation. By solving the structures of CP-loaded proteins, they revealed that this cryptic domain indeed possesses weak metalloprotease activity to cleave CP polymers into short primers and thereby to accelerate the over-all biosynthesis of CP, when starting from an un-primed situation. The data are very clearly presented and convincing control experiments have been performed. Therefore, concerning the experimental part, I have no critique.

I have only a few points for consideration:

Introduction: CP is not only a nitrogen-reserve for nitrogen-fixing cyanobacteria as stated in the first paragraph of the introduction. More recently, it was shown to enhance the efficiency of nitrogen assimilation in non-diazotrophic strains (Watzler et al., 2018). Furthermore, the references to the attempts to increase yield of CP production is not up to date. Strongly CP overproducing *Synechocystis* strain has been described in the meantime.

The second issue concerns the interpretation/discussion of the results, in particular with respect to the function of the primer-dependent enzymes. The authors argue that “all CphA1 enzymes possess very low levels of primer-independent activity for the first steps”, but that only those, which can cleave the initial CP chain (via the cryptic metalloprotease activity) are able to efficiently synthesize the polymer, because only then primers are generated that allow for an efficient continuation of CP synthesis. In this respect, there seems to be a contradiction in the argument between the in vitro and in vivo data: in vitro, the “primer-dependent” CphA1 variants didn't show any enzymatic reaction (see Fig 1 and 2), while in vitro, recombinant strains that harbour the primer-dependent variants produced appreciable amounts of CP (30 – 50 % of the level of strains with primer-independent CphA1 variants (Table 1)). This needs explanation. The explanation that the enzyme assay is not sensitive enough to detect the supposed initiating low CP synthesis by the N-domains is not convincing. How is it possible that the primer dependent enzyme produces such high amounts of CP in vivo? Is it possible that endogenous peptides in the host cell could serve as primers? Another intriguing issue: why did nature evolve primer-dependent variants of the cryptic N-domain at all? One possibility could be that this allows for an

additional level of regulation of CP synthesis by modulating the primer formation activity in this domain....

Reviewer #2 (Remarks to the Author):

The work by Schmeing and colleagues describes the elegant characterization of how an enzyme involved in the production of a naturally occurring polymer can overcome a requirement for a primer that is prevalent across other members of the same class. Notably, homologs that are primer independent utilized a segment of its own polypeptide to generate a suitable primer from existing segments of the polymer. Specifically, the N-terminus contains a metallohydrolase site that can produce suitable primers. The biochemical results are bolstered through some cryoEM analysis, along with structure-function studies that support the assignment of metallohydrolase function.

The work is elegantly done and the manuscript is concisely written. I strongly advocated it for consideration for the broad readership Nature Comm. However, there is one major point that the authors should address in greater detail before the manuscript can be published. The authors state that ICP-MS analysis is consistent with the zinc bound the the metallohydrolase active site but such analysis are prone to incorrect and artifactual results. The metal-dependency requires a more rigorous approach. At the very least, the authors ought to look at the activity of SuCphA1 in the presence of various divalent metals. It may very well be the case that zinc is the native metal but a slightly more rigorous analysis is needed to rule out alternatives.

I look forward to reviewing the revised manuscript for Nature Comm.

Reviewer #3 (Remarks to the Author):

This is another brilliant paper from the Schmeing lab - extremely convincing, very well written and presented in a way that greatly helps a general audience to understand this complex system. There is a significant amount of data included in this manuscript - a deal of it in the SI - and the results are really important. I only have some very minor comments but otherwise have no issues at all with the quality of the work being highly suitable for Nature Communications.

Of all the results, I am not sure about the value of the final table, at least as far as it is articulated in the manuscript so far - would it be better to also include a percentage of cell mass column to allow this to be compared also? This data to me seems not hugely convincing compared to the rest of the work, although perhaps I am not understanding the significance of it correctly.

The question of balance between degradation and production of cyanophycin by the primer-independent pathway is a very interesting one - I wasn't quite clear from the discussion if there was an idea of the rates of production vs degradation at relevant concentrations of different components and how these might also then change as these relative concentrations also change...could be worth discussing further.

I'm also curious if there is any evidence that the protease site is somehow "tuned-down" to prevent excessive degradation of the polymer? The discussion states that the sites seem to be very similar to others that are clearly very active - have the authors assessed if the affinity for the metal ion varies between these types of proteases (active vs those that are needed to be less so)?

This system also reminds me of one P450 where there is a moonlighting terpene synthase site within the P450 fold (10.1074/jbc.M109.064683) - although in this case it does so via modification of the enzyme fold rather than an additional domain.

Point-by-point response to reviewer comments

We thank the three reviewers for their enthusiastic reception to our manuscript. It's not often one gets such complementary reviews – they are much appreciated!

R1.0: The paper „A cryptic third active site in cyanophycin synthetase creates primers for polymerization“ by Sharon et al is excellent work...

A: Thank you for the kind words!

R1.1: Introduction: CP is not only a nitrogen-reserve for nitrogen-fixing cyanobacteria as stated in the first paragraph of the introduction. More recently, it was shown to enhance the efficiency of nitrogen assimilation in non-diazotrophic strains (Watzler et al., 2018). Furthermore, the references to the attempts to increase yield of CP production is not up to date. Strongly CP overproducing *Synechocystis* strain has been described in the meantime.

A: Apologies for the omission. We have now added the point about enhancement the efficiency of nitrogen assimilation in non-diazotrophic strains and updated the references to cite both studies mentioned.

R1.2: The second issue concerns the interpretation/discussion of the results, in particular with respect to the function of the primer-dependent enzymes. The authors argue that “all CphA1 enzymes possess very low levels of primer-independent activity for the first steps”, but that only those, which can cleave the initial CP chain (via the cryptic metalloprotease activity) are able to efficiently synthesize the polymer, because only then primers are generated that allow for an efficient continuation of CP synthesis. In this respect, there seems to be a contradiction in the argument between the *in vitro* and *in vivo* data: *in vitro*, the “primer-dependent” CphA1 variants didn't show any enzymatic reaction (see Fig 1 and 2), while *in vivo*, recombinant strains that harbour the primer-dependent variants produced appreciable amounts of CP (30 – 50 % of the level of strains with primer-independent CphA1 variants (Table 1)). This needs explanation. The explanation that the enzyme assay is not sensitive enough to detect the supposed initiating low CP synthesis by the N-domains is not convincing. How is it possible that the primer dependent enzyme produces such high amounts of CP *in vivo*? Is it possible that endogenous peptides in the host cell could serve as primers? Another intriguing issue: why did nature evolve primer-dependent variants of the cryptic N-domain at all? One possibility could be that this allows for an additional level of regulation of CP synthesis by modulating the primer formation activity in this domain....

A: We disagree that the explanation that the enzyme assay is not sensitive enough to detect the initiating low CP synthesis by the N-domains is unconvincing. We feel that the *in trans* N domain experiment definitively proves that *TmCphA* does have low CP synthesis, since the excised *Su* N domain cannot create cyanophycin itself, but imparts detectable cyanophycin synthesis by *TmCphA*. There is no other source of primer in these *in vitro* experiments other than the minute amount of cyanophycin made by *TmCphA* and cleaved by *Su* N domain.

In vivo, some other molecule would indeed act as a primer. The paper now cited as reference 23

showed that several biomolecules could prime cyanophycin synthesis, albeit with low efficiency. Peptides might also work, as R1 suggests. These alternative priming molecules, plus remnants of previous rounds of cyanophycin degradation, appears sufficient to prime enough cyanophycin production for the bacteria with non-active N domain CphA1s, or the evolutionary pressure would have ensured only active N domain containing CphA1s persist. The idea of an additional level of regulation is intriguing, and we have considered this and other modes of additional regulation, but did not feel we had sufficient data to include a discussion of such regulation.

R2.0: The work by Schmeing and colleagues describes the elegant characterization of how an enzyme involved in the production of a naturally occurring polymer can overcome a requirement for a primer that is prevalent across other members of the same class....

A: Thank you for your very positive reception to our manuscript!

R2.1: there is one major point that the authors should address in greater detail before the manuscript can be published. The authors state that ICP-MS analysis is consistent with the zinc bound the metallohydrolase active site but such analysis are prone to incorrect and artifactual results. The metal-dependency requires a more rigorous approach. At the very least, the authors ought to look at the activity of SuCphA1 in the presence of various divalent metals. It may very well be the case that zinc is the native metal but a slightly more rigorous analysis is needed to rule out alternatives.

A: We have performed additional experiments to address the issue of metal identity:

Despite the predicted difficulty in displacing the bound ion, we added a series of different metals to the *Su*CphA1 cyanophycin synthesis assays. However, exogenous metals decreased both primer-free and primer dependent synthesis, likely because of interference with the synthetic active sites.

We then made extensive attempts to strip the metal ion from the *Su*CphA1 N domain, so various metals could be assessed. Incubation of *Su*CphA with 100 mM EDTA and 1 mM 1,10-phenanthroline for a week was required to reduce primer-free synthesis to near zero. However, as soon as the chelators are removed, *Su*CphA activity returns, presumable from the N domain scavenging trace metal present in our solutions, despite our effort not to have any. Thus, we have not been able to more rigorously test other metals.

Also, we have now analyzed the metal composition of the (primer dependent) *Tm*CphA1 enzyme as an additional control. We observe that there are orders of magnitude less zinc in that sample, which is expressed and purified in the same way as *Su*CphA1 (see updated Supplementary Table 2). We took reasonable precautions when preparing the samples, such as avoiding glassware and using metal-free water.

We feel that more characterization of the metal identity is beyond the scope of this paper: Analogous characterization of peptide deformylase (PDF) enzymes are the subject of entire manuscripts. There too, attempts to strip the metal from PDF following purification are very challenging (Ragusa, Blanquet & Meinel, JMB, 1998) and can require mutations of active site

residues (Meinzel et al, FEBS Lett, 1996). PDFs are normally also purified with zinc bound (Meinzel & Blanquet, J Bacteriology 1993) and isolation of PDF bound to other metals requires their introduction during expression (Rajagopalan, Grimme & Pei, Biochemistry 2000) or purification (Ragusa, Blanquet & Meinzel, JMB, 1998) at concentrations that are unlikely to represent normal *in vivo* conditions.

We will mention the differences in Zn levels between *Su*CphA1 and *Tm*CphA1 samples in the text, and change our description of the assignment of the metal as zinc to a “tentative assignment”.

R3.0: This is another brilliant paper from the Schmeing lab - extremely convincing, very well written and presented in a way that greatly helps a general audience to understand this complex system....

A: That’s lovely to hear!

R3.1: Of all the results, I am not sure about the value of the final table, at least as far as it is articulated in the manuscript so far - would it be better to also include a percentage of cell mass column to allow this to be compared also? This data to me seems not hugely convincing compared to the rest of the work, although perhaps I am not understanding the significance of it correctly.

A: We added a column indicating the percentage of cell mass as requested. The significance of the data is to demonstrate that ability to generate primer really can be a limiting factor for cyanophycin production in *E. coli*, and likely in other heterologous hosts that have been used by the community.

R3.2: The question of balance between degradation and production of cyanophycin by the primer-independent pathway is a very interesting one - I wasn’t quite clear from the discussion if there was an idea of the rates of production vs degradation at relevant concentrations of different components and how these might also then change as these relative concentrations also change...could be worth discussing further.

A: We were somewhat hesitant to discuss relative rates because our *in vitro* measured activity cannot be precisely representative of *in vitro* rates across various cellular conditions. We have now added the following text to the discussion: “The maximum rates we observe *in vitro* indicate that polymerization of cyanophycin is several fold faster than hydrolytic cleavage, but it is difficult to relate these rates to the situation *in vivo*, where cellular conditions are not constant and availability of cyanophycin chains will change as molecules aggregate into granules. However, the accumulation of large amounts of cyanophycin in native bacteria and heterologous hosts clearly indicates the relative rates of polymerization and hydrolytic primer production are well tuned for cyanophycin biosynthesis.”

R3.3: I’m also curious if there is any evidence that the protease site is somehow “tuned-down” to prevent excessive degradation of the polymer? The discussion states that the sites seem to be very similar to others that are clearly very active - have the authors assessed if the affinity for the

metal ion varies between these types of proteases (active vs those that are needed to be less so)?

A: The affinity of the metal ion seems very high (see response to point R2.1, above). We rather believe that geometry could temper rates. See P11: “Other features of the N domain active site which may temper rate of hydrolysis are the lack of a residue for transition state stabilization (inverted zinc metallopeptidase] have a Tyr; peptide deformylase have a Gln), or the lack of an active site residue accepting a hydrogen bond from N_δ of H79. This would promote the N_ε lone-pair electrons facing the metal binding site which can be important for activity. H83 has such an interaction, but it is common for it to be seen for both active site histidines.”

R3.4: This system also reminds me of one P450 where there is a moonlighting terpene synthase site within the P450 fold (10.1074/jbc.M109.064683) - although in this case it does so via modification of the enzyme fold rather than an additional domain.

A: We were not aware of this interesting case. We agree it is worth citing and have now done so.

REVIEWERS' COMMENTS

Reviewer #1 (Remarks to the Author):

I am satisfied with the revision

Reviewer #2 (Remarks to the Author):

The revised manuscript from Schmeing and colleagues have addressed many of the (minor) concerns that were raised during the initial review. Although the authors were not able to conclusively rule out other metals ions, this was not due to lack of valiant efforts. Hence, I agree that metal analysis is beyond the scope of the current work and the assignment of zinc as tentative is sufficient. It would be of value for the authors to include all of the experiments that they did try in the SI as this would inform others interested in metalloenzyme chemistry.

I look forward to seeing the final manuscript in Nature Comm.

Reviewer #3 (Remarks to the Author):

In my opinion, the authors have made an excellent attempt to either address or discuss the comments of the reviewers. The manuscript is of very high quality and I have no hesitation in endorsing the manuscript for publication.

Point-by-point response to reviewer comments

R2: The revised manuscript from Schmeing and colleagues have addressed many of the (minor) concerns that were raised during the initial review. Although the authors were not able to conclusively rule out other metals ions, this was not due to lack of valiant efforts. Hence, I agree that metal analysis is beyond the scope of the current work and the assignment of zinc as tentative is sufficient. It would be of value for the authors to include all of the experiments that they did try in the SI as this would inform others interested in metalloenzyme chemistry.

A: Thank you for the recognition of our extensive efforts. We spent many weeks attempting to find conditions that would allow metal stripping. As described in the response to the original reviews, every time we removed the chelator, the activity returned. Because these were experiments probing different conditions for stripping, etc, we did not perform these experiments in triplicate with the same exact protocol. It would take several weeks to repeat the experiments in triplicate to allow us to add them to the supplemental information. We feel this is disproportionate, considering they are essentially failed experiments. We believe that R2's phrasing of "it would be of value" is a suggestion, rather than a requirement for publication.